# How much of the Mexican agricultural supply is produced by small farms, and how?

**María-José Ibarrola-Rivas**[1] *, **Quetzalcóatl Orozco-Ramírez**[2], **Louise Guibrunet**[3]

**1** Physics Geography Department, Geography Institute, Autonomous National University of Mexico (UNAM), Mexico City, Mexico, **2** Academic Unit campus Oaxaca, Geography Institute, Autonomous National University of Mexico (UNAM), Oaxaca, México, **3** Social Geography Department, Geography Institute, Autonomous National University of Mexico (UNAM), Mexico City, Mexico

* ibarrola@igg.unam.mx

## Abstract

The contribution of small farms to the global food supply is in debate due to lack of empirical evidence. In Mexico, small farms have been relatively important for national food supply due to an agrarian reform in the first half of the 20th century, but their role has been decreasing in the last decades. The aim of this study is to quantify how much small farms produce of the Mexican agricultural supply, and with which farming practices, using the 2019 National Agricultural Survey. The results show that small farms produce 19% of the national agricultural production with similar farming practices to those of medium and large farms. When considering imports and exports, small farms produce 15% of the national agricultural supply. The production of small farms consists mainly of cash crops (e.g. sugar cane, fruits & vegetables, animal products, fodder crops) and, to a lesser extent, staple crops such as maize and beans. The fact that small farms produce one fifth of the national production after decades of governmental support towards large farms suggests that they have resilient production systems. The results of this study support that stronger efforts should be made to enhance the role of small farms in achieving Mexican food sovereignty. This will not only have benefits in terms of food supply but may also have a wide range of social and environmental benefits.

## Introduction

Attention to the contribution of small farmers to global food supply and to agricultural sustainability is increasing [1–4]. Small farms have been associated with beneficial agricultural practices such as seed saving [5], with an agricultural production that relies on little to no external inputs, and with generating livelihoods and reducing poverty [2, 6]. These characteristics of small farms imply that they contribute to the environmental sustainability of the global food system [1, 4] and to food sovereignty [7].

Small farms represent 90% of all farms worldwide and provide direct or indirect livelihoods to up to 2.5 billion people [1]. Yet, their contribution to the global food supply remains contested. Many documents have claimed that small farms produce 70% to 80% of the global food supply (though this claim is often unsupported by empirical evidence) [8]. For instance, a

**Data Availability Statement:** All relevant data are within the paper and its Supporting information files.

**Funding:** This project was partly funded by two PAPIIT (Programa de Apoyo a Proyectos de

Investigación e Innovación Tecnológica) projects of the National Autonomous University of Mexico with projects numbers: IN300322 for Ibarrola-Rivas and IA300222 for Guibrunet. https://dgapa.unam.mx/index.php/impulso-a-la-investigacion/papiit The funders had no role in study design, data collection and analysis, decision to publish, or preparation of the manuscript.

**Competing interests:** The authors have declared that no competing interests exist.

recent report states that small farms "provide over 80 per cent of the food consumed in a large part of the developing world" and "the bulk of food in developing countries" [4 p.6, 10]. The report cites two sources of evidence to support these claims, one of which is an unpublished speech to the World Agricultural Heritage Forum by its founder and president (also former FAO senior employee) [9]. The other is an opinion piece by the president of the International Fund for Agricultural Development [10], where the same figure is stated without any references. This statement can therefore not be traced back to any evidence or empirical study that can support this claim.

In the face of this unsupported claim that small farms produce most of the global food supply, recent studies have used robust datasets and analytical methods to provide estimates of the share of small farms' production in food supply globally and in different regions. For Ricciardi et al. [8], small farms (<2ha) produce approximately one third of the global food supply. A similar figure is found in a recent FAO report, which specifies that small farms (<2ha) produce 36% of the world's food [11]. Unsurprisingly, these estimates also vary regionally, and depending on the definition that is used to characterise small farms: for Herrero et al. [12], very small farms (<2ha) contribute to about 30% of the supply of "most food commodities" in Sub-Saharan Africa, Southeast Asia, and South Asia. Samberg et al. [13 p.1], on the other hand, argue that small farms (<5ha) in Latin America, sub-Saharan Africa, and South and East Asia "are responsible for more than half of the food calories produced globally, as well as more than half of global production of several major food crops". A common conclusion of these studies is that further research is needed to quantify the role of small farms to the food supply of different regions.

To partially address this research gap, in this study, we quantify the contribution of small farms to Mexico's food supply. Mexico has a large diversity of farms in terms of size, socioeconomic, cultural, technological, and production characteristics [14–17]. Mexican studies on rurality have traditionally distinguished two large categories of farms: peasants and business farms [15, 16, 18]. Peasants (or small farmers) are usually defined as those that farm less than 2 ha or 5 ha of cropland, and as having specific characteristics such as consuming part of their production, relying on family labour, using few or no external inputs (such as inorganic fertilisers or pesticides), and having a low income [15, 16, 18].

The reality of Mexican farms is more complex than this categorisation suggests: a large spectrum of farms exists beyond peasants and business farms, and criteria for categorizing farms vary according to different authors. Schetjman [16] defined four categories of farms: peasants (relying on family labour), transition farmers (relying on both family and salaried labour, but mainly the second), agricultural entrepreneurs (only salaried labour), and livestock companies (only animal production). For peasant farms he distinguished four categories: infra subsistence (< 4 ha), subsistence (4–8 ha), stationary (8–12 ha), and those producing a surplus to sell in the market (> 12 ha). For agricultural entrepreneurs he defined three types: small (500–1250 labour days per year) medium (1250–2500 labour days per year) and large (> 2500 labour days). For livestock companies the types were: small (< 50 cattle heads or equivalent), medium (50–300 cattle heads or equivalent), and large (>300 cattle heads or equivalent). Toledo et al. [18] used a socioecological analysis to define an index composed of nine indicators to describe a farm, or any territorial unit in terms of agriculture type. The typology moves along a scale that goes from peasant to agroindustrial agriculture. The nine criteria used in their analysis are: energy, scale, self-sufficiency, labour (family vs salaried), diversity of crops, labour productivity, energy productivity, type of knowledge used, and cosmovision. For Toledo et al. [18], farms practising rainfed agriculture on less than 5 hectares are peasant and larger farms or with irrigation no matter the size are agroindustrial. For Yunez et al. [17], a family farm is one that uses more than 50% of family labour. They also separate specialised

family farms, when more than 50% of the income comes from farming, and pluriactivity family farms, when farming income is less than 50%. For de Ita [15], the two criteria for farm classification are type of labour and size. Peasant farms only use family labour, and the size is less than 20 ha. Entrepreneur farms use salary labour and are larger than 20 ha. As we can see, criteria can vary in all these typologies.

Diverse benefits of Mexican small farms have been documented in the literature. Firstly, they contribute to national food security [19] and provide 60% of agricultural labour [14]. Yet, quantifications of the contribution of small farms to the domestic supply are lacking. The Mexican government estimates that small farms (<5ha) produce 40% of the country's food supply [20], though they do not explain the methodology used to calculate this figure. Secondly, small farms contribute to agrobiodiversity, including genetic and food diversity, as well as associated cultural diversity [14, 15, 17, 21]. Because they tend to maintain traditional agroecological practices, small farms are thought to be more resource efficient and hence more environmentally sustainable than larger farms [22]

Mexico's agrarian reform following the Mexican revolution in 1910 distributed almost half of the country's land as communal land (ejidos) to small farmers [23]. With this, land tenure was recognized to indigenous communities resulting in (1) peasant's food production as the basis of the national production, and (2) preserving traditional practices [23], unlike other Latin American countries which did not have extended agrarian reform and are characterised by unequal land distribution [24]. However, this tradition of peasant farming and small farms has been displaced, to a certain extent, by the green revolution of the second half of the twentieth century [25] and more recently by the impacts of the North American Free Trade Agreement [26]. Since the 1990's, Mexican neoliberal reforms have focused on supporting large-scale and agroindustrial agriculture, thus displacing small farmers [15]. The federal government has recently started to revert this trend, with governmental programs supporting small farmers with technical and financial resources as a double strategy to reach national food sovereignty and to alleviate rural poverty [27, 28]. This strategy could be enhanced by robust and recent studies that analyse the characteristics of small farms, including their agricultural practices and the share of their production in the Mexican food supply to discuss their potential role, and changes needed, to reach Mexican food sovereignty.

The aim of this study is to quantify how much of the agricultural supply in Mexico is produced by small farms, and to analyse the farming practices used to produce it. To do this, we answer the following research questions: (1) How much of the agricultural food products are produced in Mexico by farms of different sizes? (2) What farming practices are used by farms of different sizes? (3) How much of the agricultural supply is produced by small farmers? With this, we discuss the role of small farmers for food sovereignty, and some implications of their use of different farming practices. The insights of this paper can be used to design agricultural programs to support small farmers to reach national food sovereignty.

## Material and methods

### System description and analytical steps

The system of this paper is composed of two elements (agricultural production and agricultural supply), which are analysed in three separate steps: type of farm, farming practices, and origin of supply (Fig 1).

The Mexican agricultural production is the starting point of the analysis. It is defined as the total amount of agricultural products (crops and animal products) that Mexican farms produce (step 1). The criteria to categorise the type of farm are the area of cropland and the number of animals (Table 1), since these are the most common criteria in the literature [8, 11–13,

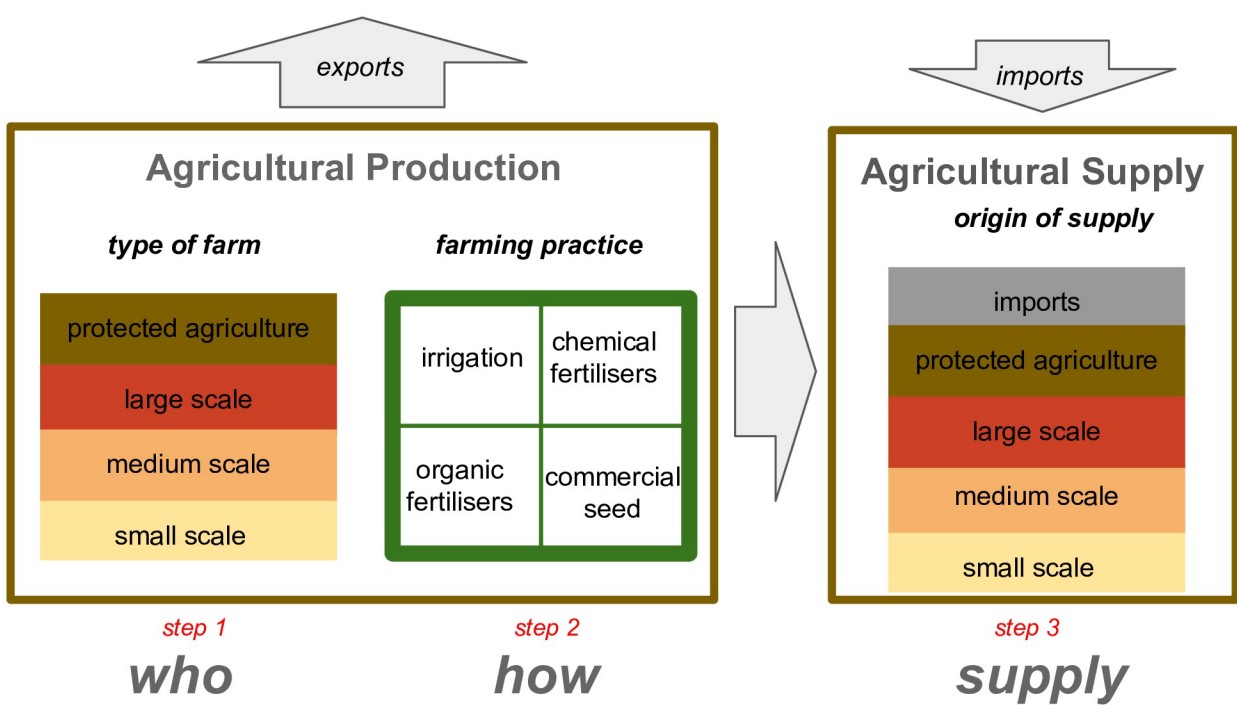

**Fig 1. System description.** Figure designed by the authors.

15, 16, 29]. Protected agriculture is considered as a different category because, compared with non-protected agriculture, its production is mainly determined by investment, technology level, and knowledge of farmers about greenhouse operation [30, 31], as well as because it can be practiced in areas unsuitable for open agriculture [30]. In addition, protected agriculture is more efficient in terms of resource use because of the use of intensive production systems with high yielding seeds and because the environmental factors can be controlled, thus reducing

**Table 1. Criteria for grouping the farms by type.** The criteria are based on the sized-based strata defined by the survey [33].

| Type of farm | Criteria for each type of agricultural products | | | |
|---|---|---|---|---|
| | Crops | Cattle products | pig products | poultry products |
| **Small scale** | 5 ha or less | ≤ 26 animals | ≤16 animals | ≤ 500 animals |
| **Middle scale** | 5 ha—50 ha | 27–120 animals | 17–150 animals | 500–20,000 animals |
| **Large scale** | > 50 ha | > 120 animals | >more than 150 animals | > 20,000 animals |
| **Protected agriculture** | Farms that use protection technologies (microtunnels or greenhouses) to produce crops. | | | |

NOTE: The survey groups cropland farms in six strata based on their cropland area, and livestock farms in four strata based on their number of animals. For crop farms, large-scale farms include strata 1 (50ha>x) and strata 2 (50ha> x <20ha); medium-scale farms include strata 3 (20ha> x <10ha) and strata 4 (10ha> x <5ha); and small-scale farms include strata 5 (5ha> x <2ha); and strata 6 (x<2). For cattle and pig farms, large-scale farms include strata 1 (120 cows >x & 150 pigs >x); mediums-scale farms include strata 2 (120 cows > x < 36 cows & 150 pigs> x <16 pigs); and small-scale farms include strata 3 (36 cows> x < 11 cows & 16 pigs> x < 3 pigs) and strata 4 (x<11 cows & x<2 pigs).

risks of losses, and enabling crop production all year round [32]. Then, we analysed the farming practices used by Mexican farms (step 2). Four indicators were used to define the farming practices: use of irrigation, use of inorganic fertilisers, use of organic fertilisers, and use of commercial seeds.

Finally, we calculated how much of the national agricultural supply (including imports and excluding exports) is produced by small farmers in Mexico (step 3). We use the concept of domestic supply defined by the FAO which includes the primary commodities produced in the country, minus exports, plus imports, plus changes in stock [34]. For this study, the changes in stock are not considered. Note that the domestic supply includes food products such as agricultural products (crops and animal products) and processed foods (e.g. sugars, alcoholic beverages, and vegetable oils), and also non-food products (e.g. crop's products for pharmaceutics & textiles), feed products, commodities for processed production, and losses in the food chain. This, food supply includes all primary production available in the country as well as the losses in the food chain, to define the national agricultural supply (Fig 1).

## Materials: Data source

The data source is the 2019 Mexican National Survey of Agriculture, produced periodically by the National Institute of Statistics and Geography (INEGI) which is the federal authority for statistics and census in Mexico [33]. We use the microdata of the survey (a database including the individual responses of each respondent). This dataset is not publicly available to ensure confidentiality for the respondents. Access to process and analyse the microdata was granted by the National Institute of Statistics and Geography (INEGI) through its Laboratory of Microdata for research purposes.

There are 3,783,588 farms in Mexico, which represents the universe to which this survey is applied [35]. A probabilistic, stratified sample scheme was applied to achieve a confidence level of 95% and a relative error of 9% (assuming a non-response rate of 30%). The initial sample was of 69,124; and responses were collected from 63,855 farms, which is a statistically representative sample for each of the food products and types of farms. Sampling weights were included in the original database [35]. The data was collected during October and November 2019 by staff trained by INEGI specifically for this survey. The staff visited each farmer's house and interviewed them filling in the online questionnaire [chapter 5 in 35]. For crop farms, data is collected for one agricultural year (1st October 2018 until 30th September 2019), to match sowing cycles. For livestock farms, data is collected for two dates (31st March 2019 and 30th September 2019) from which a yearly average is calculated. In the survey's sample, farms are categorised in six strata for crop farms and four strata for livestock farms based on their amount of cropland area and livestock per farm [33], see Table 1. The survey does not assign strata for poultry production; therefore the categorisation was based on expert knowledge [36].

The survey presents data for the 29 major agricultural products (Table 2). These 29 products, which are those considered in this study, account for 87% of the Mexican agricultural production and 84% of the Mexican domestic supply in 2019 (S1 Table, data from FAO [37–39]).

## Methods: Data analysis

**Crop farms.** For farms that produce crops, each respondent reports the production and cultivated area per crop, taking into account crops intended to be sold on the market, those kept as seeds for the next season, those used for feed, and those intended for home

Table 2. Agricultural products analysed in this study (categorised by food group). Source of data: authors' categorisation based on INEGI [33].

| Food group | Agricultural products |
|---|---|
| white maize | white maize |
| other cereals | wheat, rice, amaranth |
| bean | bean |
| vegetables | squash, onion, chilli pepper, tomato |
| fruits | avocado, strawberry, lemon, mango, apple, orange, banana, grape |
| animal feed | yellow maize, alfalfa, sorghum, soybean |
| animal products | milk, cattle meat, pork, chicken meat, eggs |
| stimulants | cocoa, coffee |
| sugar cane | sugar cane |

consumption. Farms with open agriculture (unlike those practising protected agriculture) report the share of their production that is dedicated to each use; this data is presented in S2 Table.

Using the sampling weights, these values are aggregated by crop and type of farm using the *"collapse (sum)"* function of STATA software. The results are shown in Table 3.

**Livestock farms.** As part of the survey, each respondent reported the number of animals on the farm and its production purpose (e.g. beef cows, laying hens, milking cows). We used this data, and its associated sampling weights to estimate the annual animal products production in each type of farm as follows.

The total number of animals per type of farm was calculated with the *"collapse (sum)"* function of STATA software by aggregating each type of animal reported by type of farm using the

Table 3. Distribution of farms, production, and cropland among the type of farms, and productivity indicators. Source: calculations by the authors using INEGI [33]. See S4 Table for data by food item and S1 Data for the values with all decimals.

| | | Small scale | | Medium scale | | Large scale | | Protected agriculture | | Total | |
|---|---|---|---|---|---|---|---|---|---|---|---|
| Number of Farms [Million farms] | Crops | 1.97 | 57% | 1.05 | 30% | 0.43 | 12% | 0.02 | 1% | 3.46 | 100% |
| | Cattle | 1.09 | 78% | 0.27 | 19% | 0.04 | 3% | N.A. | N.A. | 1.40 | 100% |
| | Pig | 0.34 | 17% | 0.04 | 2% | 1.60 | 81% | N.A. | N.A. | 1.98 | 100% |
| | Poultry | N.A. | 96% | N.A. | 0% | N.A. | 4% | N.A. | N.A. | N.A. | 100% |
| Production [Million tonnes] | Crops | 23.27 | 20% | 41.77 | 36% | 48.86 | 42% | 2.63 | 2% | 116.53 | 100% |
| | Cattle | 3.09 | 21% | 3.84 | 26% | 7.59 | 52% | N.A. | N.A. | 14.52 | 100% |
| | Pig | 0.13 | 8% | 0.13 | 8% | 1.34 | 84% | N.A. | N.A. | 1.60 | 100% |
| | Poultry | 0.00 | 0.0% | 0.00 | 0.1% | 6.42 | 99.9% | N.A. | N.A. | 6.43 | 100% |
| Cropland area [Mha] | Crops | 3.27 | 21% | 5.15 | 33% | 7.24 | 46% | 0.05 | 0.3% | 15.71 | 100% |
| **Average farm Production [ton / farm]** | **Crops** | **11.83** | | **39.89** | | **114.61** | | **145.69** | | **33.69** | |
| | **Cattle** | **2.83** | | **14.43** | | **199.59** | | **N.A.** | | **10.38** | |
| | **Pig** | **0.39** | | **3.36** | | **0.84** | | **N.A.** | | **0.81** | |
| | **Poultry** | **N.A.** | | **N.A.** | | **N.A.** | | **N.A.** | | **N.A.** | |
| **Average yield [ton/ha]** | **Crops** | **7.12** | | **8.11** | | **6.75** | | **50.86** | | **7.42** | |
| | **Cattle** | **N.A.** | | **N.A.** | | **N.A.** | | **N.A.** | | **N.A.** | |
| | **Pig** | **N.A.** | | **N.A.** | | **N.A.** | | **N.A.** | | **N.A.** | |
| | **Poultry** | **N.A.** | | **N.A.** | | **N.A.** | | **N.A.** | | **N.A.** | |

sampling weights. Having the total animals in each type of farm, we estimated the share of animals in each farm (e.g. x% of national milking cows are in small-scale farms). This percentage was multiplied by the national annual production of beef, milk, pork, chicken meat, and eggs, respectively, reported by the Agriculture and Fishery Information System of Mexico in 2019 [40]. With this, we estimated the amount of each animal product produced by each type of farm. The results are shown in Table 3.

For beef production, we included beef cows and half of the "dual" animals (milk and beef producing cows). For milk production, we included the milking cows and half of the "dual" animals. For pig production, we considered all pigs reported by each farm (any age). The sample for poultry production farms is a deterministic sample which does not come with sampling weights, which means that the total national values cannot be quantified. To address this, we used the distribution of these farms among the small-, medium-, and large- scale farms to estimate the national distribution of these farms. The sample of the poultry production farms represent 90% of the national production of eggs and chicken meat (source: personal communication with the staff of the National Agricultural Surveys of INEGI). To estimate the poultry meat production, we considered growing chickens, and for egg production, we considered the hens reported by each farm.

**Farming practices.** Each respondent reported whether they used an irrigated or rainfed system, applied inorganic fertilizers and/or organic fertilizers, and the type of seed used. The organic fertilizers used by farmers are mainly manure of different livestock, compost, and other homemade fertilizers such as earthworm humus [33]. For the type of seed, respondents could choose from four categories: (1) landrace or farm saved seeds, (2) improved (hybrid varieties or varieties that have been selected by a rigorous method to assure quality, performance, and yield if environmental conditions are optimal), (3) certified or registered, (4) genetically modified. We grouped the improved, certified or registered, and the genetically modified seeds as "commercial seeds", and the landrace or farm saved seeds as "non-commercial seeds". These data were used to define four variables characterizing farming practices: (a) use of irrigation, (b) use of inorganic fertilizers, (c) use of organic fertilizers, (d) use of commercial seeds. Farming practices of protected agriculture are not reported in the survey, so these farms were not considered in the analysis. This analysis also excluded animal products; but by including fodder crops, we considered these products indirectly. The answers of all respondents were aggregated per type of farm and type of farming practice, using sampling weights, with the *"collapse (sum)"* function of STATA software. The results are shown in Fig 2.

**Agricultural supply.** Finally, to calculate the agricultural supply, we quantified the exports and imports of the 29 agricultural products included in this study (S3 Table). The exports were subtracted equally for all types of farms from the national production, and the imports were added as a new category making up the domestic supply (Fig 1, step 3). With this, we calculated how much of the agricultural supply is produced by small farms, farms of other sizes, or obtained from imports. See S3 Table for details.

## Results

### Agricultural production in Mexico by type of farm

Most of the food produced in Mexico (64%) comes from large farms (Table 3). Small farms produce 19%, medium-scale farms produce 33%, and protected agriculture produce 2% of the Mexican agricultural production. The share of the production of each type of agricultural product is also different among the four types of farms (S4 Table). Almost half of the white maize (the main food product in the Mexican diet) is produced by large-scale farms, and 17% is produced by small farms. Beans, also one of the main food products of the Mexican diet, are

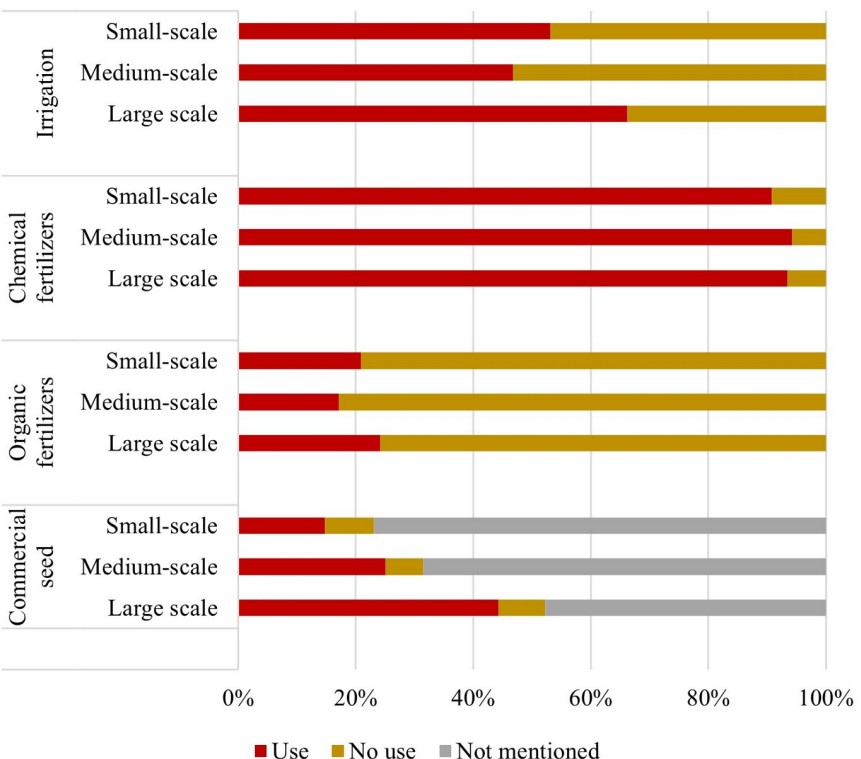

**Fig 2. Farming practices used in the production of crops per type of farm.** Source of data: calculations by the authors using the National Agricultural Survey [33]. The data plotted in this graph can be found in the S1 Data.

mainly produced by large-scale farms, followed by medium farms, while small farms produce 6% of the national production. Vegetables are mainly produced by large-scale farms, followed by protected agriculture farms (the latter producing almost exclusively vegetables). Animal products and fodder crops are also mostly produced by large-scale farms (68% and 62%), followed by medium-scale farms (18% and 27%), and by small-scale farms (14% and 16%).

Even though small-scale farms produce a relatively small share of the national production, they account for the largest share of farms (57% of crop farms, 78% of cattle farms, 89% of pig farms, and 96% of poultry farms) (Table 3). This difference in the number of farms of different sizes and their total production can be explained by the fact that the average production per farm is much higher in large scale farms, followed by medium- and small- scale farms (see 4th row in Table 3). Large-scale farms' average production is ten times higher than small-scale farms for crops, 70 times higher for cattle farms, and 2000 times higher for pig farms. However, the average crop yield is similar among the three types of farms (Table 3). Protected agriculture, on the other hand, boasts yields ten times higher than all other farms—this can be explained by the fact that it produces almost exclusively vegetables, which yields are higher than for other types of crops (such as cereals).

The production values presented in Table 3 include products that are sold by farmers as well as those intended for other uses (animal feed, seeds, and home consumption). Only 6%, 4% and 9% of the production is not sold by farmers, for small-, medium and large-scale farms respectively (S4 Table). This percentage diverges among food crops. For instance, less than 90% of maize, beans, and alfalfa are sold—across all farm types. This means that the three types of farms use a similar share of their production of these crops for food, feed and seeds.

## Farming practices used in the production

Farming practices are similar across the three types of farms (Fig 2). Over 90% of farms of any size use inorganic fertilizers. Less than a quarter of farms (of any size) use organic fertilizers. This means that most crops are produced with inorganic fertilizers and without organic fertilizers across all farm sizes. The use of irrigation systems shows a different pattern. It is mostly used by large-scale farms (66%) but is also commonly used by small- and medium-scale farms (53% and 47% respectively).

Most farms failed to report the type of seeds they use (see grey values in Fig 2). Only 23%, 32% and 52% of the small-, medium, and large-scale farms, respectively, reported the type of seed used. Therefore, more data is needed to discuss the distribution of the type of seeds among small-, medium-, and large-scale farms.

## How much of the agricultural supply is produced by small farms?

7.8 million tons out of the 139 million tons of the agricultural products produced in Mexico are exported, which accounts for 6% of the national production (S3 Table). These are mainly vegetables (2.6 million tonnes) and fruits (2.2 million tonnes). The imports of the agricultural products considered in this study represent 36.95 million tons per year and are mainly animal feed (26.8 million tons: mainly yellow maize) and other cereals for food (mainly wheat which amounts to 5.5 million tons). By adding the imports and subtracting the exports from the national agricultural production, we obtain the national supply of agricultural products, which amounts to 168 million tons in 2019. 22% of the supply originates from imports, 15% is produced by Mexican small-scale farms, 26% by Mexican medium-scale farms, 36% by large-scale farms, and only 1% by protected agriculture (Fig 3, Table 4, S3 Table).

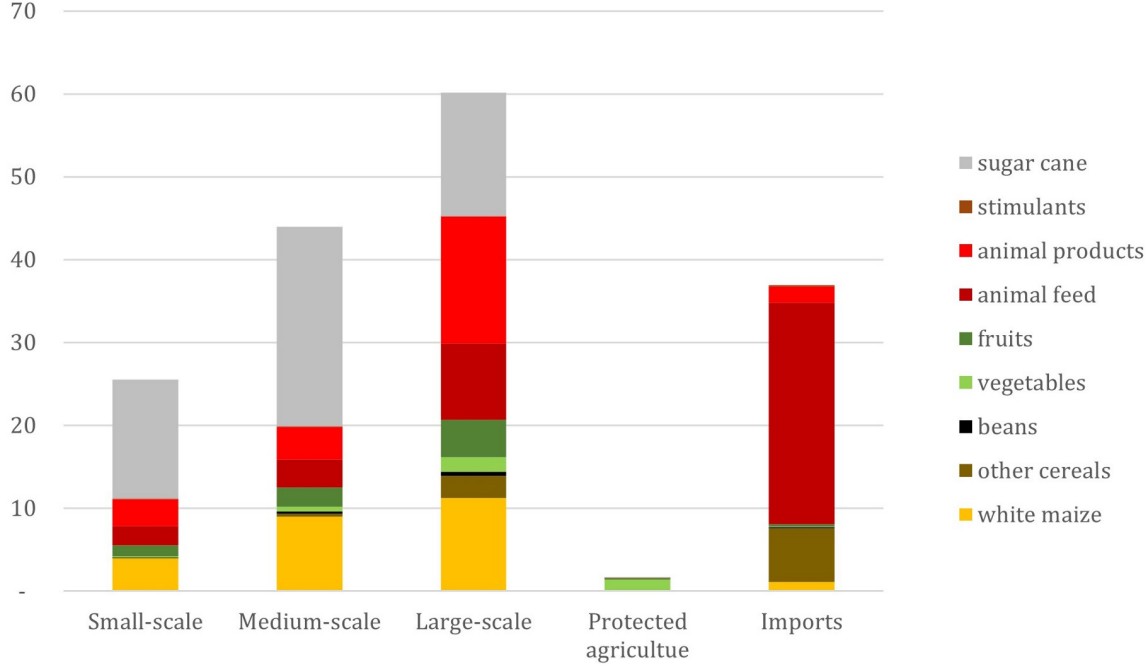

**Fig 3. Source of the Mexican domestic supply in million tons, by food group.** Source of data: calculations by the authors using the National Agricultural Survey [33] and FAO [38]. The data plotted in this graph can be found in S1 Data.

**Table 4. Source of national agricultural supply for each food product million tons and share of the total supply by food product (in italics).** Source of data: calculations by the authors using the National Agricultural Survey [33] and FAO [37]. The details on the values with decimal points are shown in the S1 Data.

| Food group | Agricultural Product | Small scale | Medium scale | Large scale | Protected agriculture | Imports | Total |
|---|---|---|---|---|---|---|---|
| white maize | white maize grain | 3.99 | 9.02 | 11.23 | 0.00 | 1.12 | 25.36 |
| | | *16%* | *36%* | *44%* | *0%* | *4%* | *100%* |
| other cereals | wheat | 0.02 | 0.29 | 2.59 | - | 5.53 | 8.42 |
| | | *0.2%* | *3%* | *31%* | *0%* | *66%* | *100%* |
| | rice | 0.01 | 0.04 | 0.10 | - | 0.94 | 1.08 |
| | | *1%* | *3%* | *10%* | *0%* | *86%* | *100%* |
| | amaranth | 0.004 | 0.001 | 0.001 | 0.000 | - | 0.01 |
| | | *68%* | *20%* | *11%* | *1%* | *0%* | *100%* |
| | **TOTAL** | **0.03** | **0.32** | **2.69** | **0.00** | **6.47** | **9.51** |
| | | ***0%*** | ***4%*** | ***28%*** | ***0%*** | ***68%*** | ***100%*** |
| bean | bean | 0.05 | 0.25 | 0.48 | 0.00 | 0.12 | 0.90 |
| | | *5%* | *28%* | *54%* | *0%* | *13%* | *100%* |
| vegetables | squash | 0.02 | 0.03 | 0.08 | 0.00 | - | 0.13 |
| | | *17%* | *20%* | *62%* | *1%* | *0%* | *100%* |
| | onion | 0.03 | 0.10 | 0.63 | 0.00 | 0.09 | 0.85 |
| | | *4%* | *12%* | *74%* | *0%* | *11%* | *100%* |
| | chilli pepper | 0.03 | 0.41 | 0.54 | 0.36 | 0.00 | 1.34 |
| | | *3%* | *30%* | *40%* | *27%* | *0%* | *100%* |
| | tomato | 0.03 | 0.07 | 0.51 | 1.01 | 0.00 | 1.62 |
| | | *2%* | *4%* | *31%* | *63%* | *0%* | *100%* |
| | **TOTAL** | **0.12** | **0.60** | **1.75** | **1.37** | **0.09** | **3.94** |
| | | ***3%*** | ***15%*** | ***45%*** | ***35%*** | ***2%*** | ***100%*** |
| fruits | avocado | 0.22 | 0.41 | 0.37 | - | - | 1.01 |
| | | *22%* | *41%* | *37%* | *0%* | *0%* | *100%* |
| | strawberry | 0.01 | 0.01 | 0.02 | 0.16 | 0.00 | 0.20 |
| | | *4%* | *6%* | *9%* | *79%* | *2%* | *100%* |
| | lemon | 0.09 | 0.22 | 0.65 | 0.00 | 0.00 | 0.97 |
| | | *10%* | *23%* | *67%* | *0%* | *0%* | *100%* |
| | mango | 0.15 | 0.21 | 0.39 | 0.00 | 0.00 | 0.76 |
| | | *20%* | *28%* | *52%* | *0%* | *0%* | *100%* |
| | apple | 0.02 | 0.15 | 0.22 | 0.09 | 0.16 | 0.64 |
| | | *3%* | *23%* | *35%* | *14%* | *25%* | *100%* |
| | orange | 0.64 | 1.22 | 1.60 | 0.00 | 0.02 | 3.48 |
| | | *18%* | *35%* | *46%* | *0%* | *1%* | *100%* |
| | banana | 0.22 | 0.11 | 1.14 | 0.00 | 0.00 | 1.47 |
| | | *15%* | *7%* | *78%* | *0%* | *0%* | *100%* |
| | grape | 0.01 | 0.01 | 0.15 | 0.00 | 0.06 | 0.23 |
| | | *3%* | *5%* | *65%* | *1%* | *26%* | *100%* |
| | **TOTAL** | **1.36** | **2.34** | **4.55** | **0.25** | **0.25** | **8.75** |
| | | ***15%*** | ***27%*** | ***52%*** | ***3%*** | ***3%*** | ***100%*** |

*(Continued)*

**Table 4.** (Continued)

| Food group | Agricultural Product | Small scale | Medium scale | Large scale | Protected agriculture | Imports | Total |
|---|---|---|---|---|---|---|---|
| animal feed | yellow maize grain | 0.43 | 0.93 | 3.98 | - | 19.70 | 25.05 |
| | | 2% | 4% | 16% | 0% | 78% | 100% |
| | alfalfa | 1.55 | 1.50 | 2.56 | 0.00 | - | 5.61 |
| | | 28% | 27% | 45% | 0% | 0% | 100% |
| | sorghum grain | 0.35 | 0.93 | 2.39 | - | 0.63 | 4.30 |
| | | 8% | 21% | 56% | 0% | 15% | 100% |
| | soybeans | 0.00 | 0.03 | 0.27 | 0.00 | 6.48 | 6.78 |
| | | 0% | 1% | 4% | 0% | 95% | 100% |
| | **TOTAL** | **2.34** | **3.39** | **9.20** | **0.00** | **26.80** | **41.74** |
| | | **6%** | **8%** | **22%** | **0%** | **64%** | **100%** |
| animal products | milk | 2.57 | 3.28 | 6.64 | - | 0.21 | 12.70 |
| | | 20% | 26% | 52% | 0% | 2% | 100% |
| | beef | 0.50 | 0.53 | 0.90 | - | 0.00 | 1.93 |
| | | 26% | 27% | 47% | 0% | 0% | 100% |
| | pork | 0.13 | 0.13 | 1.34 | - | 0.70 | 2.30 |
| | | 6% | 5% | 58% | 0% | 31% | 100% |
| | chicken meat | 0.00 | 0.00 | 3.47 | - | 0.97 | 4.45 |
| | | 0% | 0% | 78% | 0% | 22% | 100% |
| | eggs | 0.00 | 0.00 | 2.95 | - | 0.05 | 3.00 |
| | | 0% | 0% | 98% | 0% | 2% | 100% |
| | **TOTAL** | **3.20** | **3.94** | **15.30** | **-** | **1.94** | **24.38** |
| | | **13%** | **16%** | **63%** | **0%** | **8%** | **100%** |
| stimulants | cocoa | 0.01 | 0.01 | 0.00 | - | 0.04 | 0.06 |
| | | 23% | 10% | 5% | 0% | 62% | 100% |
| | coffee | 0.09 | 0.06 | 0.04 | 0.00 | 0.12 | 0.31 |
| | | 29% | 20% | 13% | 1% | 37% | 100% |
| | **TOTAL** | **0.11** | **0.07** | **0.05** | **0.00** | **0.16** | **0.38** |
| | | **28%** | **18%** | **12%** | **1%** | **41%** | **100%** |
| sugar cane | | 14.35 | 24.00 | 14.90 | 0.00 | - | 53.25 |
| | | 27% | 45% | 28% | 0% | 0% | 100% |
| **Total agricultural supply** | | **25.56** | **43.94** | **60.15** | **1.63** | **36.95** | **168.22** |
| | | **15%** | **26%** | **36%** | **1%** | **22%** | **100%** |

Different food products are sourced in different ways (Table 4). For instance, wheat, rice, animal feed, and cocoa are the food categories where imports represent more than half of the domestic supply. For these categories, the difference between the production of small-, medium-, and large-scale farms appears to be relatively less important when imports are considered. Only 0.2%, 1%, and 6% of the domestic supply of wheat, rice, and animal feed, respectively, is produced by small farmers.

The food products that are most largely produced by small farms (at least 20% of the domestic supply), in comparison with the other farms or imports, are amaranth (68%), avocado (22%), mango (20%), alfalfa (28%), milk (20%), beef (26%), sugar cane (27%), cocoa (22%), and coffee (29%). Al these products are considered "cash crops" or "commercial crops", animal products or animal feed. None of them are staple foods. Also, production by small-scale farms never exceeds 30% of the national supply, apart from amaranth, but its production accounts to only 5.4 thousand tonnes in 2019, which is only 0.003% of the total agricultural supply.

## Discussion and conclusion

Mexican small farms, characterised in our study as those that have less than 5 hectares of cropland, 16 pigs or less, 26 cows or less, or 500 chickens or less, produce 19% of Mexico's agricultural production, and 15% of Mexico's agricultural supply. The reduction in the percentage from production to supply is explained by the large quantities of food imports. These values differ from the widely reported statement that small farmers produce most of the food globally, and the Mexican government's statement that small farmers produce 40% of the Mexican food [20]. Recent studies [8, 13] estimating the global production of small farms have already shown a smaller estimate than the unsupported figure of 70%-80%. These authors estimate that small farms (under 5 hectares) produce 44%-48% [8] and 55% [13] of global food supply. Still, their results are much higher than the estimates of this study. This can partly be explained by the fact that small farms are more common to Africa and Asia than to Latin America [13]. This can also illustrate how Mexican small farms and the peasant farming traditions which had been preserved thanks to the agrarian reform of the early 20th century are losing importance as a result of the wave of neoliberal reforms of the last three decades [15].

Similarly to Ricciardi et al. [8], our results show that farms of different sizes tend to produce different food products. Ricciardi et al. show that small farms produce mainly staple food crops (cereals, roots and tubers, pulses) and fruits, and large-scale farms produce oil crops, vegetables, and other crops. However, we found a different pattern for the Mexican case. The main staple food crops for Mexican diets (white maize and beans) are mainly produced by large-scale farms. In contrast, the agricultural products that small farms most produce are "commercial" or "cash crops": coffee and cocoa, sugar cane, followed by animal products, fruits, animal feed, and white maize; with only 16% of white maize being produced by small-scale farms (Table 4).

Further, farming practices do not differ widely according to the farm size (Fig 2). The farming practices of small-, medium-, and large- scale farms follow a similar pattern: most of the food is produced using inorganic fertilizers, around half of farms of any size use irrigation, and a low share of the production relies on organic fertilizers. However, these findings must be corroborated and expanded through further research. In this analysis, we only considered a dichotomous characterisation of farming practices (whether farms use a certain agricultural input or not). But differences may exist in the amount or degree of use among the farms. For instance, small farms may use a smaller amount of fertilizer and irrigation than large scale farms. These differences must be quantified in a further study. In addition, the use of non-commercial seed has several social, cultural and agrobiodiversity implications. Efforts have been made to preserve diversity in Mexican seeds, particularly for native maize varieties which are preserved mostly by small farmers [41]. Unfortunately, as most farms fail to report the type of seed they use, our study does not allow us to draw conclusions regarding the prevalence of the use of commercial seeds in Mexican farms. Further research is hence needed to identify the type of seeds used in Mexican agricultural production to enable discussions on the related agrobiodiversity, cultural and political implications of the use of different seeds.

Thus, our results show that Mexican small farms are not the main contributors to the Mexican agricultural supply, and their farming practices may have similar environmental implications than medium- and large- scale farms. However, these findings do not diminish the relevance of small farms for Mexico's food sovereignty and employment. Small farms represent the majority of farms in Mexico, a fifth of the cropland area, and the main provider of rural livelihoods. Small farms use marginal lands which otherwise would not be used by large-scale farms. In addition, they are a space for the generation of traditional knowledge and cultural heritage, which is a key tool for the resilience and adaptability of the national food system [42,

43]. Recently, some efforts have been made by the Mexican government to support small farmers [27], however they have not been enough. Since the 1990's, the governmental agricultural programs for small farmers have focused on increasing family income, rather than on enhancing production capacity [17]. The fact that small farms persist with little governmental support suggests that their production system is resilient due to their social land tenure [15], to their culture and cosmovision [18, 44, 45], and to their economic strategies [44]. According to some authors [2, 6] small farms have the potential to increase their food productivity increased while reducing poverty.

To increase the relevance of small farms in Mexican food sovereignty, stronger efforts are needed to increase their role in the production of the main staple and nutritious food for the Mexican population, which will not only have benefits in terms of enhancing food sovereignty, but also may have a wide range of social and environmental benefits. This can be the focus of further agricultural programs promoting the production of nutritious staple food such as beans and white maize.

## Supporting information

**S1 Table. Agricultural production and domestic supply of the 29 agricultural products used in this study.**
(DOCX)

**S2 Table. Total production of no market and market production produced by each type of farm and each type of crop.**
(DOCX)

**S3 Table. Imports and exports of agricultural products estimated for this study.**
(DOCX)

**S4 Table. Distribution of agricultural products production by type of farm in Mexico.**
(DOCX)

**S1 Data. Data of the tables and figure shown in the main article and in the S1–S4 Tables.**
(XLSX)

## Acknowledgments

The data used in this paper were processed and analysed under project number LM-530 at the Microdata Laboratory of the National Institute of Statistics and Geography (INEGI) at their offices in Mexico City. The authors give special thanks to the staff of INEGI Microdata Laboratory (María de Lourdes Garrido Blancas, Lidia Hernández Hernández, Liliana Martínez Matias, Andrea López Ortiz, and Natalia Eugenia Volkow Fernández) and the experts of the agricultural census and survey at INEGI (Victor Dueñas de la Rosa, Karina Lara Fernández, Luis Fernando Esteves Cano, Ismael Saucedo Silva) for their unconditional support during our work at the Microdata Laboratory, even in times of pandemic.

## Author Contributions

**Conceptualization:** María-José Ibarrola-Rivas, Quetzalcóatl Orozco-Ramírez, Louise Guibrunet.

**Formal analysis:** María-José Ibarrola-Rivas, Quetzalcóatl Orozco-Ramírez.

**Methodology:** María-José Ibarrola-Rivas, Quetzalcóatl Orozco-Ramírez.

**Visualization:** María-José Ibarrola-Rivas.

**Writing – original draft:** María-José Ibarrola-Rivas, Louise Guibrunet.

**Writing – review & editing:** Quetzalcóatl Orozco-Ramírez, Louise Guibrunet.

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
