## [Decision Letter · Decision Letter 0]

6 Jul 2023

PONE-D-23-16291How much of the Mexican agricultural supply is produced by small farms, and how?PLOS ONE

Dear Dr. Ibarrola-Rivas,

Thank you for submitting your manuscript to PLOS ONE. After careful consideration, we feel that it has merit but does not fully meet PLOS ONE’s publication criteria as it currently stands. Therefore, we invite you to submit a revised version of the manuscript that addresses the points raised during the review process. Please submit your revised manuscript by Aug 20 2023 11:59PM. If you will need more time than this to complete your revisions, please reply to this message or contact the journal office at plosone@plos.org. Please include the following items when submitting your revised manuscript:A rebuttal letter that responds to each point raised by the academic editor and reviewer(s). You should upload this letter as a separate file labeled 'Response to Reviewers'.A marked-up copy of your manuscript that highlights changes made to the original version. You should upload this as a separate file labeled 'Revised Manuscript with Track Changes'.An unmarked version of your revised paper without tracked changes. You should upload this as a separate file labeled 'Manuscript'.

We look forward to receiving your revised manuscript.

Kind regards,

Noé Aguilar-Rivera

Academic Editor

PLOS ONE

Journal Requirements:

3. We note that you have referenced (unpublished) on page 3 which has currently not yet been accepted for publication. Please remove this from your References and amend this to state in the body of your manuscript: (ie “Bewick et al. [Unpublished]”) as detailed online in our guide for authors

Reviewers' comments:

Reviewer's Responses to Questions

**Comments to the Author**

1. Is the manuscript technically sound, and do the data support the conclusions?

Reviewer #1: Yes

Reviewer #2: Partly

2. Has the statistical analysis been performed appropriately and rigorously? 

Reviewer #1: I Don't Know

Reviewer #2: No

3. Have the authors made all data underlying the findings in their manuscript fully available?

Reviewer #1: Yes

Reviewer #2: No

4. Is the manuscript presented in an intelligible fashion and written in standard English?

Reviewer #1: Yes

Reviewer #2: Yes

5. Review Comments to the Author

Reviewer #1: In this manuscript (MS) the respected authors have written about the contributions of small farms to the Mexican food industry. The MS contains several citations relevant to the topic discussed. The authors cite documents supporting and opposing their theory.

Major issues: none

Minor issues:

There are some minor typoes and missing punctuation marks at the end of a sentences.

In the PDF format Figure 1 is pixelated. I'm nt sure if this is caused by the editorial system (e.g. downscaling the resolution) or the original image was less than 300 DPI in quality.

The english of the Materials and methods section could be improved.

Please correct the name of Schetjman to Schejtman.

Line 90: FAO, 2014 is not listed in references.

In line 131 the correct citation is: (Bustamante et al., 2017)

In line 137: (Sánchez, 2003)

Line 186: correctly: de Ita

In line 220 I would use beef cows or cows for meat production instead of "fattening cows". In lines 222 and 223, and later on instead of "fattening" I'd rahther use the expression "for meat production".

In line 246 instead of "recycled seeds" I'd suggest to use "farm saved seeds". Also in line 247 "improved" seeds should be rephrased to correctly specify/clarify the meaning. Line 253 use "fodder crops" instead of "feed crops". Line 271 farms in prular at the end of the sentence. Line 277 use "fodder crops" instead of "animal feed". Line 286 use "cattle" instead of "bovine".

I would suggest to explain why protected growing is more efficient than other production methods. For example in protected cultivation plants could be grown year round, and because more of the environmental factors can be controlled. Also it would be worth mentioning that the cultivation there is more intensive, and growers use high yielding cultivars/seeds.

In section 3.2 the authors write about "chemical fertilizers". I was wondering if this could be rephrased in the MS as "inorganic fertilizers" as in my opinion it describes them more accurately. Also in this section "organic fertilizers" need clarification. Does this phrase mean that these fertilizers are certified Organic by some authorities or this just menas that these fertilizers are animal byproducts (like manure) and compost?

Line 343: FAOSTAT, 2019is not listed in references.

Line 365 add a space after the comma: "cropland, 16 pigs ...".

Line 403, please omit "robust".

Line 525 correctly: de Ita, ...

Lines 421 and 540: What is the correct name of the Author, Leff or Left? Please change it both places to match.

Line 549: Lowder, S. K., Sánchez, M. V., & Bertini, R. (2021) is not cited in the text.

Reviewer #2: This is a very interesting subject matter especially for developing countries agricultural economies. My first worry with this research is the fact that it didnot take into account stable crop production. This means that important production for home consumption and gifts has been neglected somehow and therefore the results of this analysis might not be acurate. This may explain the reason why their results are much lower than the national reported estimates. I will suggest they look for a way to include this data either through a case study or so and redo the analysis.

Elswhere, some of the percentages did not sum up to 100 meaning there are some problems with the calculations

6. PLOS authors have the option to publish the peer review history of their article (what does this mean?). If published, this will include your full peer review and any attached files.

Reviewer #1: No

Reviewer #2: No

---

## [Decision Letter · Decision Letter 1]

14 Sep 2023

PONE-D-23-16291R1How much of the Mexican agricultural supply is produced by small farms, and how?PLOS ONE

Dear Dr. Ibarrola-Rivas,

Thank you for submitting your manuscript to PLOS ONE. After careful consideration, we feel that it has merit but does not fully meet PLOS ONE’s publication criteria as it currently stands. Therefore, we invite you to submit a revised version of the manuscript that addresses the points raised during the review process.

We look forward to receiving your revised manuscript.

Kind regards,

Noé Aguilar-Rivera

Academic Editor

PLOS ONE

Reviewers' comments:

Reviewer's Responses to Questions

**Comments to the Author**

1. If the authors have adequately addressed your comments raised in a previous round of review and you feel that this manuscript is now acceptable for publication, you may indicate that here to bypass the “Comments to the Author” section, enter your conflict of interest statement in the “Confidential to Editor” section, and submit your "Accept" recommendation.

Reviewer #1: All comments have been addressed

Reviewer #2: (No Response)

2. Is the manuscript technically sound, and do the data support the conclusions?

Reviewer #1: Yes

Reviewer #2: Partly

3. Has the statistical analysis been performed appropriately and rigorously? 

Reviewer #1: Yes

Reviewer #2: I Don't Know

4. Have the authors made all data underlying the findings in their manuscript fully available?

Reviewer #1: Yes

Reviewer #2: No

5. Is the manuscript presented in an intelligible fashion and written in standard English?

Reviewer #1: Yes

Reviewer #2: Yes

6. Review Comments to the Author

Reviewer #1: (No Response)

Reviewer #2: I still think this manuscript has a very important place in food security research and development in developing countries. But it is really weak with serious adjustments to be made on the data used and the methodology. See my comments in the manuscript.

7. PLOS authors have the option to publish the peer review history of their article (what does this mean?). If published, this will include your full peer review and any attached files.

Reviewer #1: No

Reviewer #2: No

---

## [Editor Report · Decision Letter 2]

25 Sep 2023

How much of the Mexican agricultural supply is produced by small farms, and how?

PONE-D-23-16291R2

Dear Dra. Maria-Jose Ibarrola Ibarrola-Rivas

We’re pleased to inform you that your manuscript has been judged scientifically suitable for publication and will be formally accepted for publication once it meets all outstanding technical requirements.

Kind regards,

Noé Aguilar-Rivera

Academic Editor

PLOS ONE
---

## [Editor Report · Acceptance letter]

27 Sep 2023

PONE-D-23-16291R2 

How much of the Mexican agricultural supply is produced by small farms, and how? 

Dear Dr. Ibarrola-Rivas:

I'm pleased to inform you that your manuscript has been deemed suitable for publication in PLOS ONE. Congratulations! Your manuscript is now with our production department. 

Kind regards, 

on behalf of

Dr. Noé Aguilar-Rivera 

Academic Editor

PLOS ONE